# Model-Free Model Reconciliation

**Sarath Sreedharan, Alberto Olmo Hernandez, Aditya Prasad Mishra** and
**Subbarao Kambhampati**

School of Computing, Informatics, and Decision Systems Engineering,
Arizona State University, Tempe, AZ 85281 USA
{ ssreedh3, aolmoher, amishr28, rao } @ asu.edu

## Abstract

Designing agents capable of explaining complex sequential decisions remains a significant open problem in human-AI interaction. Recently, there has been a lot of interest in developing approaches for generating such explanations for various decision-making paradigms. One such approach has been the idea of *explanation as model-reconciliation*. The framework hypothesizes that one of the common reasons for a user's confusion could be the mismatch between the user's model of the agent's task model and the model used by the agent to generate the decisions. While this is a general framework, most works that have been explicitly built on this explanatory philosophy have focused on classical planning settings where the model of user's knowledge is available in a declarative form. Our goal in this paper is to adapt the model reconciliation approach to a more general planning paradigm and discuss how such methods could be used when user models are no longer explicitly available. Specifically, we present a simple and easy to learn labeling model that can help an explainer decide what information could help achieve model reconciliation between the user and the agent with in the context of planning with MDPs.

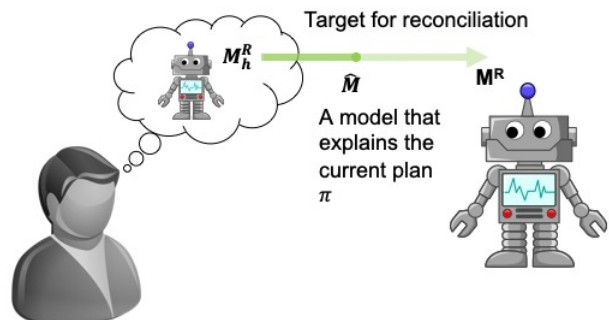

Figure 1: A general overview of the explanation as model reconciliation.

## 1 Introduction

A significant barrier to integrating AI systems into our daily lives has been their inability to interact and work with us humans in an intuitive and explicable manner. Orchestrating such interactions would require the agents to have the ability to help users in the loop better understand the rationale behind their various actions. Thankfully there has been a lot of effort within the AI research community to develop systems capable of holding explanatory dialogues with users and thus help them understand the decisions under question [David W. Aha and Magazzeni, 2018; Daniele Magazzeni, 2018]. Such explanatory systems could help users resolve confusions regarding agent decisions that may stem from either a (1) lack of understanding (or even misunderstanding) of the task or (2) from their inferential limitations. While many earlier works in explanation have generally focused on the latter (cf. [Khan *et al.*, 2009; Hayes and Shah, 2017; Seegebarth *et al.*, 2012; Topin and Veloso, 2019]), there is a growing consensus on the importance of explanatory mechanisms that can help bridge the knowledge asymmetry between the system and the user.

In particular, in **explanation as model-reconciliation** [Chakraborti *et al.*, 2017] we had studied the problem of reconciling knowledge asymmetries between the user and the agent within the context of planning. Works in this direction have generally looked at cases where the user's model of the task (i.e their belief about the initial state, the transition dynamics, and the goal) is known beforehand (in a representation scheme comparable to the one used by the agent) and do not match the agent's model. This mismatch means that the user would not be able to correctly evaluate the validity or the optimality of a given plan. Thus in this paradigm the explanations consist of information about the agent's model that the user could incorporate into his/her own model to correctly evaluate the plan in question.

Unfortunately, it is not always possible to have access to such models. In the most general case, we are dealing with the user's model of the agent and hence the user may not be capable of presenting traces or decisions that could be generated from this model. Even if the system tries to learn such a representation based on interactions with the user, there is no guarantee that the specific representational assumptions of the learned model and the vocabulary used would be satisfied by the user's mental model.

The definition of explanation as model reconciliation may leave one with the idea that there is no way around it. How could one ever truly perform effective reconciliation when there exists no user model guiding us to choose the parts of the model, which when revealed to the user will help them correctly evaluate the current decision? Are we left with revealing the entire agent model to the user as the only option? In this paper, we propose a simple and intuitive way we could still generate minimal explanations in the absence of declarative models. We argue that we could drive such explanations by using learned models that can predict how human expectations could be affected by possible explanations (derived completely from information about the agent model) and in fact show how this method could be viewed as a variation of previous approaches that have been put forth to identify explicable behavior.

We will start by extending model reconciliation to the more general setting of planning with Markov Decision Processes (Section 4). The rest of the paper will investigate how these ideas could be used when the human mental model of the task is unavailable, and will formulate a learning problem that allows us to learn simple models that could be used to identify minimal explanations (Section 5). Finally, we will evaluate our method on a set of standard MDP benchmarks and perform user studies to validate its viability (Section 6).

## 2 Background

Figure 1 presents a general schematic representation for explanation as model reconciliation. The automated agent (henceforth referred to as robot) starts with a model $\mathcal{M}^R$ that can be used to generate a decision $\pi$ (where depending on the context, $\pi$ may be a single action, plan, policy or a program). In this setting, $\mathcal{M}_h^R$ corresponds to the human's preconceived notions about the robot model. The explainer's job then becomes providing information about the model $\mathcal{M}^R$, such that the updated human model can correctly evaluate the validity of the robot decisions.

In this case, the robot could have chosen to provide the entire model, but for most realistic tasks, such models could be quite large, so dumping the entire model could be both unnecessary and impractical. It's also well known that people generally prefer explanations that are selective [Miller, 2018; Lombrozo, 2006]. Thus the users would be happier with explanations that asks them to update a subset of beliefs as opposed to a complete update. Note that $\mathcal{M}^R$ need not be the original agent model, but rather some abstraction/approximation of the underlying robot model (that conserves some desired property of the decision like optimality or validity). In [Chakraborti et al., 2017] where model reconciliation was first introduced, $\mathcal{M}^R$ was a classical planning model hence inherently interpretable and thus the reconciliation could easily be achieved, but the idea could be applied beyond just planning models. For example, one could understand the explanation methodology used by LIME [Ribeiro et al., 2016] as being a special case of model reconciliation. In their case, they assume the human model is empty and $\mathcal{M}^R$ is an approximation of the underlying decision model is automatically generated for each decision using a set of prede-

fined features.

In this work, we will be looking at the agents that use discounted infinite horizon Markov Decision Processes (or MDPs) [Russell and Norvig, 2003] as the decision making framework. Each MDP $\mathcal{M}$ is defined by a tuple $\langle S, A, T, R, \gamma, \mu \rangle$, where the $S$ provides the set of possible atomic states, $A$ defines the set of actions, $T$ is the transition function, $R$ the reward, $\gamma$ the discounting factor (where $0 \leq \gamma < 1$) and $\mu$ corresponds to the distribution of possible initial states. $T : S \times A \times S \rightarrow [0,1]$ provides the probability that for given state $s \in S$, the execution of an action $a$ would induce a transition to a new state $s'$, and $R : S \times A \times S \rightarrow \mathbb{R}$ defines the reward corresponding to this transition. The solution concept in MDP takes the form of a policy $\pi$ that maps each state to a potential action. A policy is said to be optimal for $\mathcal{M}$ (denoted as $\pi_{\mathcal{M}}^*$) if there exists no other policy that dominates the given policy in terms of the expected value of states (where the value of a state $s$ under a policy $\pi$ for a model $\mathcal{M}$ is denoted as $V_{\mathcal{M}}^\pi(s)$). Executing the policy in a state results in a sequence of state action state tuples called *execution trajectory* or simply a *trajectory*, denoted as $\tau = \langle (s_1, a_1, s_2), ..., (s_{n-1}, a_{n-1}, s_n) \rangle$ and we will use $P_{\mathcal{M}}(\tau|\pi)$ to denote the probability of sampling the given trajectory $\tau$ for a policy $\pi$ in model $\mathcal{M}$.

In the explanatory setting we are interested in, the robot uses a model $\mathcal{M}^R = \langle S, A, T^R, R^R, \gamma^R \rangle$ of the task to come up with the policy to act on. For now we will assume this MDP already defines an interpretable model and the human uses a model $\mathcal{M}_h^R = \langle S, A, T_h^R, R_h^R, \gamma_h^R \rangle$ to evaluate it (we will relax this assumption in later sections). Now the task ahead of us will be to formulate how we could still identify minimal information that could resolve user confusion when $\mathcal{M}_h^R$, but before we can do that we need to reinterpret the ideas of inexplicability and the idea of model reconciliation that was defined in [Chakraborti et al., 2017] in the context of MDPs and we will start by considering a simple scenario.

## 3 Illustrative Example

Consider a warehouse scenario, where a robot is tasked with moving packages from racks and dropping them off at the dispatch chute. The robot is powered by a battery pack that can be recharged by visiting its docking station. The docking station also doubles as a quality assurance station that the robot needs to visit whenever it picks up a box labeled #013 (which means the box is fragile). The robot's operations are mostly deterministic, apart from a small probability of slipping (0.25) in some cells, that could leave the robot in the same position.

Now suppose the warehouse has just hired a new part-time employee to oversee the operations. The employee is just getting used to this new setting and is puzzled by the robot's decision to once in a while take a detour from the drop-off activity and visit a specific position of the factory floor (which is, in fact, the docking location). If we wished the robot to be explainable, then it would need to be capable of helping the employee better understand the underlying model used by the robot (i.e achieve some form of model reconciliation). Given the fact that the robot may not have an exact model of the user,

one way to achieve this could be by providing robot's entire model to the user. Unfortunately, this could easily overwhelm the user.

Another possibility could be to allow the user to specify which robot actions appear inexplicable, and focus on providing facts relevant to those actions. This explanation may still prove to be quite verbose and may in fact not help resolve their confusion. For example, imagine a case where the robot is visiting the station to recharge its batteries and the human says that the visit action is inexplicable. Now even if the robot mentions that visiting the station recharges it, the employee may still be confused if they are under the incorrect assumption that the robot is operating on full battery. Similarly, if the human had expected the robot to go to the docking station due to some confusion regarding the box codes, the human may mark the robot decision to not go to the dropoff as being inexplicable and the explanations that could resolve the confusion may have little to do with that specific action marked as inexplicable.

## 4 Explanation as Model Reconciliation For MDPs

In this setting the human and robot models are captured as MDPs defined over the same set of states and thus we can characterize both models by the tuple $\theta = \langle \theta_T, \theta_R, \theta_\gamma, \theta_\mu \rangle$, where the $\theta_T$ provides the set of parameters that defines the transition probabilities $P(.|s,a)$, while $\theta_R$ the parameters corresponding to the reward function, $\theta_\gamma$ the parameters corresponding to the discount factor and $\theta_\mu$ the parameters for the initial state distribution. For simple MDP models with atomic states, $\theta_T$ contains parameters of the categorical distribution for each transition ($\theta_\mu$ will contain similar parameters for the initial state distribution), $\theta_R$ contains the reward associated with each transition (an $\langle s, a, s' \rangle$ tuple) and $\theta_\gamma$ just contains the value of the discount factor. The specific instantiations of the parameters for each model $\mathcal{M}$ is captured as $\theta(\mathcal{M})$. For simplicity, we will denote each of the unique parameters in the tuple $\theta$ using indexes. For example, $\theta_T^{s,a}(\mathcal{M}^R)$, will correspond to the parameters for the distribution $P(.|s,a)$ for the model $\mathcal{M}^R$.

If we use $\mathbb{M}$ to capture the set of all possible models and $\Theta = \theta_T \times \theta_R \times \theta_\gamma \times \theta_\mu$, then model reconciliation operation can be captured as a function $\mathcal{E}_{\langle \mathcal{M}_h^R, \mathcal{M}^R \rangle} : 2^\Theta \to \mathbb{M}$ that takes in a set of model parameters and generates a new version of the model $\mathcal{M}_h^R$ where the set of specified parameters will be set to values from $\mathcal{M}^R$. For example, $\hat{\mathcal{M}} = \mathcal{E}_{\langle \mathcal{M}_h^R, \mathcal{M}^R \rangle}(\theta_T^{s_1,a})$ will be a new model such that $\theta(\hat{\mathcal{M}})$ will be identical to $\theta(\mathcal{M}_h^R)$, except that $\theta_{T_{\hat{\mathcal{M}}}}^{s_1,a}$, will be equal to $\theta_{T^R}^{s_1,a}$.

Practically, the model reconciliation operation corresponds to the robot informing the human about some part of its model. This communication could incur cost and we can capture this by using the cost function $\mathcal{C} : 2^\Theta \to \mathbb{R}$ that maps a given set of a threshold to a cost.

Now the question we need to ask is whether the agent is trying to explain its policy or if it is trying to explain some behavior (i.e an execution trace). Most of the earlier work that looks at model reconciliation explanation (cf. [Chakraborti *et al.*, 2017; Sreedharan *et al.*, 2018a; 2018b]) has looked at sequential plans and has generally ignored this differentiation and treated the problem of explaining plans to be same as that of explaining behavior. In general, a given plan or policy compactly represents a set of possible behaviors and the choice of explaining behavior *vs* explaining the plans/policies could affect the content of the explanation being given. For example, when explaining policies there is the additional challenge of presenting the entire policy to the user and the explainer may need to justify action choices for extremely unlikely states or contingencies. On the other hand, when explaining a given set of behaviors the explainer needs to only justify their action choices for cases they actually witnessed. For example, when explaining traces from the warehouse scenario, given the small probability of slipping, the robot may never have to mention what to do when it slips, but on the other hand if we are dealing with full policies, the agent may need to talk about the states where the robot is in the slipped positions and they need to get up from that position and move on.

Explaining policies or plans becomes more relevant when we consider explanatory dialogues where the agent and the user are trying to jointly come to agreement on what policy/plans to follow (eg: decision support systems), while the latter may be more useful when the user is observing some agent operating in an environment.

With respect to policies, we assume that the user is presented with the entire policy. A given policy is said to be explicable to the human, if the policy is optimal for the human model. Therefore the goal of the explainer becomes that of ensuring the optimality of the given policy

**Definition 1.** *A set of parameters $\theta_\mathcal{E}$ corresponds to a **complete policy explanation** for the given robot policy $\pi_{\mathcal{M}^R}^*$, if the policy is also optimal for $\mathcal{E}_{\langle \mathcal{M}_h^R, \mathcal{M}^R \rangle}(\theta_E)$ and is said to be the minimally complete policy explanation if there exists no other complete explanation $\theta_{\mathcal{E}'}$, such that, $\mathcal{C}(\theta_{\mathcal{E}'}) < \mathcal{C}(\theta_\mathcal{E})$*

Finding a complete policy explanation is relatively straightforward (the set of all parameters automatically meets this requirement). The more challenging case becomes that of finding the minimal or the cheapest explanations i.e. the minimally complete explanations. Such minimally complete explanations can be calculated by adopting a search strategy similar to [Chakraborti *et al.*, 2017]. The search can start at the human model and try to find the minimal number of parameters that needs to be updated in the human model for the current policy to become optimal. Similar to generating minimally complete explanations, i.e, we can also generate monotonic explanations (i.e explanations where no further information about parameters in the robot model can affect the optimality of the current plan).

In the case of policies, we can also describe explicable planning and balancing cost of explanations with that of choosing policies that are inherently explicable, where inexplicablity score ($\mathcal{I}_\mathcal{E}$) of a policy $\pi$ is defined as

$$\mathcal{I}_\mathcal{E}(\pi, \mathcal{M}_h^R) = |E[V_{\pi*}^{\mathcal{M}_h^R}(s)|s \sim \mu_h^R] - E[V_\pi^{\mathcal{M}_h^R}(s)|s \sim \mu_h^R]|$$

Where $\pi*$ is the optimal policy in the human model. Explicable planning thus becomes the problem of choosing

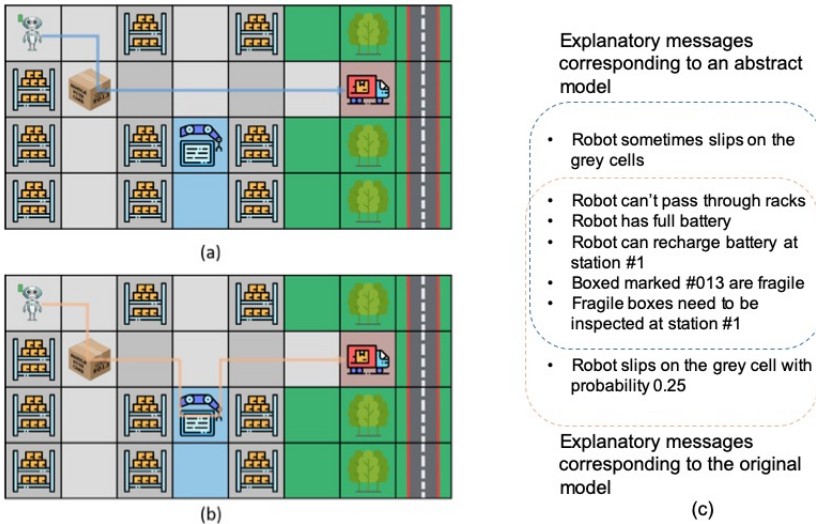

Figure 2: Subfigure (a) shows a visualization of a trajectory expected by the user described in the illustrative example, and (b) shows the visualization of a trajectory the user may observe. Subfigure (c), shows the various explanatory messages that could be used in this scenario, note that the messages span information from multiple abstractions of the given task

policies that minimize inexplicability score [Kulkarni *et al.*, 2016], while minimizing the potential loss in optimality due to the policy choice (since the most explicable plan may not be an optimal policy). Balanced planning, as studied in [Chakraborti *et al.*, 2019; Sreedharan *et al.*, 2019], proposes going one step further and also takes into account possible savings in inexplicability score that can be achieved by providing explanation (while incurring additional cost of communicating the required explanations).

For explaining behavior, we will look at the simplest case, namely the agent needs to explain a set of behaviors that the user has just observed. We will assume that the observer has full observability of the state and is seeing the robot behavior for the first time. In such a setting, a given trace $\tau$ would appear explicable to the user if it could be sampled from their expected MDP policy (i.e a policy optimal in their model) or more generally, i.e $P_{\mathcal{M}_h^R}(\tau|\pi) > \delta$, where $\delta$ is some small threshold. [1]

**Definition 2.** *A set of parameters $\theta_{\mathcal{E}}$ corresponds to a **complete behavior explanation** for a set of traces $\mathbb{T} = \{\tau_1, ...\tau_n\}$, if $\forall \tau \in \mathbb{T}$, $\exists \pi$ such that $P_{\mathcal{E}_{\langle \mathcal{M}_h^R, \mathcal{M}^R \rangle}(\theta_E)}(\tau|\pi) > \delta$ and $\pi$ is an optimal policy for the model $P_{\mathcal{E}_{\langle \mathcal{M}_h^R, \mathcal{M}^R \rangle}(\theta_E)}$. The explanation is said to be the minimally complete behavior explanation if there exists no other complete explanation $\theta_{\mathcal{E}'}$, such that, $\mathcal{C}(\theta_{\mathcal{E}'}) < \mathcal{C}(\theta_{\mathcal{E}})$*

Note that given the above definition, if $\delta$ is set very high it may not be possible to find a complete explanation, as the trace may genuinely contain low probability transitions. In this work we will assume $\delta$ to be zero.

While model reconciliation could be an important component of either policy or behavior explanation, the applicabil-

ity of the model reconciliation explanations on their own for policies is limited by the fact that in all but problems with the smallest state spaces, the user would have trouble going over the entire policy. Thus in these settings, explanatory systems would need to also utilize policy approximation or summarization methods, then allow users the ability to drill down on policy details as required. Since our main goals was to focus on developing approaches that allow us to generate model reconciliation explanations without explicitly defined user models, the rest of the paper will mostly focus on behavior explanation. In Section 8, we will have a brief discussion on how these methods could potentially be extended to policy explanation scenarios.

## 5 Explaining Without Explicit Human Mental Models

Now we will look at how we can identify cheap complete behavior explanations when the human model is unknown. We will go one step further from identifying not only the parameters that need to be explained, but also capturing the right modality/abstractions to present the information about the parameters. That is, we will no longer assume that the human is using a full MDP model to come up with their decisions. Instead, the robot starts with a set of explanatory messages $\Psi = \{m_1, m_2, ..., m_n\}$ that can be presented to the user. Where the messages correspond to a set of parameter values (the parameters corresponding to a set of messages $\{m_1, ..m_k\}$ is denoted as $\mathcal{E}(\{m_1, ..m_k\})$) of the model as captured in some abstraction of this model and has a corresponding cost ( $\mathcal{C}$ ) associated with it. The abstractions to consider may depend on the specific scenario and the previous information about the intended users (laypeople vs. experts). Some simple possibilities may be to consider qualitative models (say non-deterministic ones instead of stochastic)

---
[1] We use $\delta$ in the general case to allow for the possibility that people can be surprised by unlikely events of non-zero probability

and considering state abstractions the given task. Note that, technically $\mathcal{E}(\Psi)$, need not span the set of all possible model parameters, but could rather be limited to a subset of parameters identified to be relevant to the given problem. One possible way may be to consider variations of explanation techniques like MSE [Khan *et al.*, 2009] to identify set of possible factors that affect the optimality of each action. In Figure 2, the subfigure (c) shows a set of possible explanatory messages for the warehouse domain, that consists of each parameter mapped to some english statement. For models captured using factored representations that use relational or propositional fluents, such statements could be easily generated using templates (cf. [Hayes and Shah, 2017]).

Given this setting, we will now make some simplifying assumptions, namely, (1) the order in which the explanatory messages are presented does not matter (2) we have access to a set of observers with similar models and they share this model with the target user (3) the robot is viewing the task at the same level or at a more detailed level of granularity than the user and (4) the user and robot have some shared vocabulary in regards to the task. While assumption (1) is easily met since we are mostly dealing with model information and (4) is a prerequisite for most explanatory approaches, in section 8 we will discuss how we can possibly relax requirements (2) and (3).

Now our goal is to learn a predictive model that is able to predict whether a given user would find a given $\langle s, a, s \rangle$ tuple explicable and how the user's perception changes with the given explanatory messages.

For example, at the beginning of an episode the user may be presented with the following explanatory messages, $\hat{\Psi} = \{m_1 = \text{"Robot slips with probability 0.25 at grey cells"}\}$, which corresponds to the fact that $P(s_i|a, s_i) = 0.25$, for all states $s_i$ where the feature grey cell is true and for all actions $a$. Now the user will be presented with a sequence of transitions, say $\langle (1,2), \text{right}, (2,2) \rangle$ and asked whether the transition was explicable or not. Then the tuple $\langle \langle (1,2), \text{right}, (2,2) \rangle, \{m_1\}, l_1 \rangle$, where $l_1$ is the label assigned by the user to the transition, becomes input to our learning method.

The exact function that we would want to learn would be

$$\mathcal{L}(\langle s, a, s' \rangle, \{m_1, ..., m_k\}) = \begin{cases} 1 & \text{if } \langle s, a, s' \rangle \sim \\ & \pi^*_{\mathcal{E}_{\langle \mathcal{M}_h^R, \mathcal{M}^R \rangle}(\theta(\{m_1, ..., m_k\})}(s) \\ 0 & \text{otherwise} \end{cases}$$

Note that this is a modified version of the sequential model we introduced in [Zhang *et al.*, 2017] for identifying whether a given plan is explicable or not. Though our methods vary in some significant aspects, namely, (1) we allow for the possibility that the explicability of the actions/traces could be affected by explanations provided by the system; (2) we no longer use labels of high level tasks as a proxy for the explicability of the trace. Instead, we just use a simple binary label on whether the transition is explicable or not; (3) we no longer consider sequence models but rather a much simpler labeling model that maps a single transition to the explicability label. We argue that in cases where the human is markovian on the same set of features as the agent, this rather simpler model

suffices.

It is also important that our learning approach is more tractable than the ones studied in [Zhang *et al.*, 2017], since in their case to build a balanced dataset (of explicable and inexplicable plans), they would need to uniformly sample through the entire plan space (an extremely hard endeavour with no obvious known approaches), while we stick to traces generated from the optimal policy and only need to randomly generate possible sets of explanatory messages, which is clearly a smaller set.

Once we have learned an approximation of the above labeling function $\hat{\mathcal{L}}$, the problem of explanation generation for a trace $\tau = \langle s_0, a_0, s_1, ..., s_n, a_n, s_{n+1} \rangle$ becomes that of finding the subset of $\Psi$ that balances the cost of communication with the reduction in the inexplicability of the given trace, i.e

$$\arg\min_{\hat{\Psi}} (\mathcal{C}^{\mathcal{M}}(\hat{\Psi}) + \alpha * \Sigma_{i=0}^n (1 - \hat{\mathcal{L}}(\langle s_i, a_i, s_{i+1} \rangle, \hat{\Psi})))$$

Where $\hat{\Psi}$ is a subset of $\Psi$ and $\alpha$ is some scaling factor that balances the cost of explanation with the number of inexplicable transitions for a given trace.

## 6 Evaluation

The success of the approach described above would be directly dependent on whether we can learn high accuracy labeling models. Once we have access to such a model, we could be quite confident in our ability to generate useful explanation (provided the user's model is the same as the one the labeler was trained on) and identifying the best explanation becomes a matter of just searching for the required subset of messages that minimizes the objective defined in section 5. So to evaluate the method our focus was on identifying if it was possible to learn high accuracy models. We validated our approach on both simulations and on data collected from users.

### 6.1 Evaluation on Simulated Data

For simulations, we used a slightly modified versions of the Taxi domain [Dietterich, 1998] (of size 6*6), the Four rooms domain [Sutton *et al.*, 1999] (of size 9*9) and the warehouse scenario (of size 9*9) described before (implemented using the SimpleRL framework [Abel, 2019]). For each domain, we start with an MDP instance (henceforth referred to as the robot model) and then create a space of possible user models by identifying a set of possible values for each MDP parameter. For example, in the taxi domain the parameters include position of the passengers, their destination, the step cost, discounting etc., for the Four rooms this included the goal locations, locations with negative rewards, discounting, step cost, slip probability, etc., and finally for the warehouse, the position of the box, the position of station #1, the step cost, slipping probabilities and the discounting factors were selected as potential parameters that can be updated. In this setting, we assume that there exists a single explanatory message for each possible parameter.

For each individual test, we select a random subset of three parameters and then randomly choose a value for each of these. We then treat this new MDP model as a stand-in for

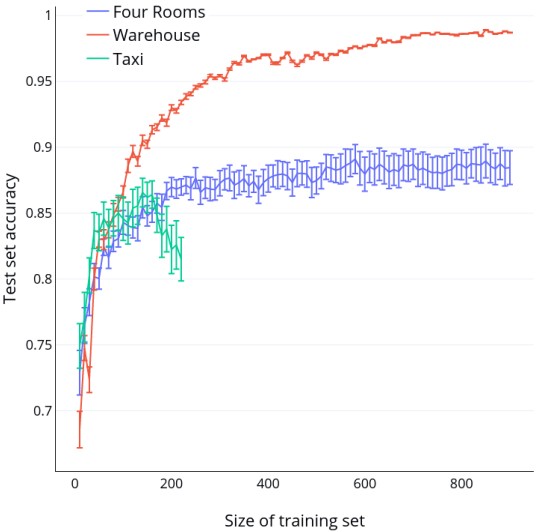

Figure 3: The test accuracy for increasing sizes of training set.

the user model and use it to label traces generated from the original MDP. The traces were generated by choosing a random initial state and then following the optimal policy of the robot until either the terminal state is reached or the trace length reaches a predefined limit. For each trace, a random subset of the explanations was selected and presented to the human. This means updating the MDP parameters to their corresponding values in the robot model only for the parameters specified by the current subset of explanation. Each individual transition was then labeled using this updated MDP. A transition was labeled as inexplicable if the action is not the optimal one in the human model (i.e. Q value is lower) or the next state had a probability of occurring of $\delta = 0$.

We then used this set of labeled transitions to create a training set and test set for a decision tree learner. The input features to the decision tree consist of current state features, (just x and y for Four rooms and the position of the the taxi and passengers for the Taxi domain and for Warehouse it included the position of the robot and the fact whether the agent picked up the box or visited station #1), the index of the action and features capturing the current subset of explanations being considered. In each Warehouse and Four rooms test instance, we collected 900 unique data points as training set and 100 data points as the test set. Due to the complexity of the taxi domain, we generated less data points (since for each different explanation subset we need to solve a new planning problem) and used close to 220 unique points as training data and on average 28 data points as the test set.

We then tested on 20 such instances for each domain. Figure 3 plots the average test accuracy for models trained with training sets of varying sizes. As evident from the graph, a simple decision tree seems to be able to easily model the effect of explanations on labeling for these simulated scenarios. We chose a simple learning model to establish the viability of this method, but one could easily see that the use of more sophisticated learning methods and/or more informed features should lead to better results.

## 6.2    User Studies

Next, we wanted to establish if we can still learn such simple models when the labels are collected from naive users. Our goal here is not to consider scenarios with possible differences in the user's knowledge, but rather cases where, even in the presence of a set of users with similar backgrounds, their responses to explanations would be too varied to learn useful models. To test this, we used the Warehouse domain as a test bed and collected feedback on how users would view the explicability of traces generated from this domain when presented with explanatory messages detailed in Figure 2.

For the study, we recruited 45 master turkers from the Amazon Mechanical Turk. Each participant was provided with the URL to a website (`https://goo.gl/Hun3ce`) where they could view and label various robot behaviors. We considered a setting where the robot had a full battery, but was picking up a fragile box and thus still needs to visit the station #1. The robot could slip on some cells marked in dark grey with probability 0.25 (slipping here meant the robot picture is tilted to give an impression that it slipped on the cell and didn't prevent the robot from moving to the next cell). To make sure that all the users had similar mental models at the start, they were provided with the following facts, (a) that robot couldn't pass through racks, (b) whenever the robot runs low on battery it needs to get to Station 1, (c) whenever the robot has a green battery sign next to the robot, that means their battery is full and (d) the robot needs to take the shortest route to the goal. Also, they were presented with an example trace in this instructions section and were made to take a small pre-test that allowed them to revise the above facts in various scenarios. After the pre-test, they were shown eight traces from the robot policy sampled according to their probabilities. After the first trace, the user was given an explanation message before each trace, where the message was taken from the seven possible messages (the order of the messages was always randomized).

From the data collected from 45 turkers, we removed data from seven users, based on the fact they didn't find any of the transition in the first trace (i.e the case where no explanation was provided) inexplicable. We imagine this number would go down when we move to expert users or users who are invested in the success of the robot. The data generated for the remaining 38 users were then used to train a decision tree. Since the placement of other objects in the environment were fixed, we were able to use rather simple features for the model like the current position of the robot (x and y), previous position (again x and y), the action, whether they have slipped and finally the explanations given. We found the model to have an average 10-fold cross validation score of 0.935. For a randomly generated train and test split (where the test split was 10% and contained around 7% inexpicable labels) the precision score was 0.9637 and the recall score was 0.9568.

Furthermore, we could see that the model was able to correctly predict the usefulness of intuitive minimal explanations for the given scenario. For example, it predicted that while the robots decision to visit station #1 would be considered inexplicable by the user in the absence of any explanation, the user would mark it as explicable when they are explained about the box being fragile and that fragile boxes need to be

inspected at station #1. In fact the model predicted that only the message that "fragile boxes need to be inspected at station #1" is enough to convince the user about the need for that action (i.e the user could deduce that the box must have been fragile). This shows that such learned models may help us generate cheaper explanations (the above set of explanations is smaller than the corresponding minimal complete behavior explanation for the domain), by taking into account the users ability to correctly predict missing information in simple cases. Another point of interest was that the model predicted all slipping events as explainable even in the absence of any explanations. The cases where the user saw a slip before being told about the possibility of slipping was rare (since there are two explanatory messages related to slipping and the probability of slipping was 0.25). Furthermore when we went over the data, we found that in most such cases, the users did mark it as explainable. This may be because the effect of slipping may not have been that detrimental to the overall plan (it doesn't take you off the current path). It would be interesting to see if this result would be the same in cases where slipping was a more likely event and if it had a more apparent effect on the robot's plan.

## 7  Related Work

To the best of our knowledge, this work represents the first attempt at learning proxies for user mental models that allows an agent to predict the potential impact of providing explanations as model reconciliation to observers. With that said, there have been works that have looked at the problem of generating explanations in the presence of model uncertainty for human models. In particular, our previous works like [Sreedharan *et al.*, 2018a; 2018b] have looked at cases where the agent has access to a set of potential human models. One drawback of considering a set of possible models is either they would need to have explicit sensing to identify the user model (which could mean asking a large number of questions to the user) or providing a large amount of information to cover the space of all possible models. In our work, the problem of identifying the specifics of the user model is resolved through an offline training process.

Another work quite related to the discussion covered in this paper is [Reddy *et al.*, 2018], wherein the authors tried to identify cases where they can learn a potential model for the human's expectation of the task transition dynamics when they do not align with the real world dynamics. Unlike their work, we do not assume that the user can provide traces for the given task, rather they may be able to provide some high-level feedback on the action (i.e. they may not be able to do or even know the right action but may be able to point out actions or transitions that surprise them). Moreover, their work requires that the user and the robot must have the same reward function, which is again an assumption we do not make. Even if we had followed their technique to learn a potential approximation of the human's transition model for the task, there is no guarantee that the learned representation would be one that makes sense to the human.

## 8  Discussion and Conclusion

This paper proposes a possible way in which model reconciliation explanation could be applied to cases where the user model is unknown. The method described here is a rather simple and general method to identify information that could potentially affect the user's mental model and produce effects that align with the agent's requirements. There is no requirement here that the messages have to align with actual facts about the world. This again points to the rather troubling similarities between the mechanisms needed to generate useful explanations and lies [Chakraborti and Kambhampati, 2018].

Two important assumptions we made throughout the work is that the user only considers the current state (as defined by the robot) to make their decisions and we have access to a model that was learned from interactions to previous users who had similar knowledge level to the current user. Relaxing the first assumption would require us to go beyond learning models that map each transitions to labels. Instead we have to consider sequential labeling models (for example models based on LSTM or CRF) of the type considered in [Zhang *et al.*, 2017] to capture the human's expectations. For example, we considered a simple extension of the warehouse domain where the human believes the robot should visit two locations (i.e the human state contains variables that record whether the user has visited the locations). Even though here the user is considering a more detailed model, we were able to learn labeling models of 80% accuracy by using simple CRFs. As for the second, instead of assuming that all users are of the same type, a more reasonable assumption may be that the users could be clustered into N groups and we could learn a different labeling model for each user type. Now we still have a challenge of identifying the user type of a new user and one way to overcome this would be by adopting a decision-theoretic approach to this problem and modeling it as a POMDP (where user labels become observations and previously learned user models the observation models).

The work discussed in this paper only covers explanations that allow the user and the system to reconcile any model difference. This only covers a part of the entire explanatory dialogue. Even if there is no difference in models, the user may still have questions about parts of the policy or may raise alternative policies they think should be followed. This may arise from a difference in inferential abilities and may require providing information that is already part of their deductive closure eg: help them understand the long term consequences of taking some actions. Once you have access to a set of such messages one could use a method similar to the one described in the paper to find the set of helpful ones. Unlike the model reconciliation setting where the messages stand for information about the model, it is not quite clear how one could automatically generate such messages.

## Acknowledgments

This research is supported in part by the ONR grants N00014-16-1-2892, N00014-18-1-2442, N00014-18-1-2840, the AFOSR grant FA9550-18-1-0067, NASA grant NNX17AD06G and a JP Morgan AI Faculty Research grant .

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
