# OpenReview forum: "Model-Free Model Reconciliation"
_icaps-conference.org/ICAPS/2019/Workshop/XAIP — XAIP 2019_

### Official Review · AnonReviewer1 · 2019-05-08
**Learning which explanatory messages to provide to people**

**Rating:** 3
**Confidence:** 3

**Review:**

This paper extends/generalises an existing explanation approach called 'model reconciliation' in two ways. First, it generalises from classical planning models to MDPs, and second, it generalises by extending to cases in which there are no mental models of the human. The general idea is that offline learning is employed to determine which elements of a set of explanatory messages is the most useful to give to people The idea is evaluated on some synthetic problems, including one in which user studies were performed that evaluated whether there is consistency between what is learnt and what questions users have. No evaluation on the explanations themselves.

Some comments for improvement:

- I think the paper could be improved by defining what question the explanations are answering here. In the context of explainable planning, the explanations are not really answering why particular actions or sequences, but are informing users of the the robot model (modulo the user model). The explanations still require the observer to do quite a bit of work; namely, the judge given the parameters whether this is optimal. So, it would be interesting to define: what question would a user have that this model would be an answer to (if I'm making sense).

- It is not quite clear (to me at least) what the purpose of the theta function/model is at the start of Section 4; that is, the combination of the two models. In the definition of \Epsilon, we have 2^{\Theta} --> \mathcal{M}, but as far as I can tell, \Theta and \mathcal{M} are the same thing and we could just use \mathcal{M}.

- While I understand where the authors are going with the 'model-free' idea, the model-free moniker and the section 5 title could be considered somewhat cheeky considering that the prevailing assumption is that there are a collection of 'training' humans that provide the necessary mental models that are learnt from.

- Section 6 could be improved by clearly outlining the goals of the experiment, as well as giving a slightly better structure to the experiment design: independent variables, measures, etc.

A question for the authors:

- The explanation process is very much tied to the idea of the explanation being optimal with respect to the human mental model. While the authors point to literature outlining that giving explanations that conform to existing beliefs is often good, is there any evidence to suggest that optimality is a criterion that people typically care about?


Minor things:
- I think both capital theta and mathcal{M} are undefined (first used in Section 4, paragraph 2 to define the epsilon model reconciliation function); but I was able to infer what they meant.

- The title of the paper is accurate, but will probably not be matched by keyword searches, etc. Perhaps adding "explainable planning" or something to the title?

- Section 5 title "without access to human mental models"; perhaps "without an explicit human mental model". We cannot access human mental models ever :)

- Lots of non-capitalised terms in the referneces; e.g. "mdp" instead of "MDP"

---

> ### Author Response · Authors · 2019-05-10
> **RE: Learning which explanatory messages to provide to people**
>
> We thank the reviewer for their comments and feedback and we will work on incorporating them into the revised version of the paper.
>
> Explanatory Questions: Most explanations generated by model reconciliation methods, can be seen as answers to the question “Why this plan?” (where there may be implicit foils). We agree that for complex domains and plans with long horizons, model reconciliation information may no longer be enough and the explainer may need to provide inferential assistance to the user to help them better understand the plan/policy in question.
>
> Parameterized representation of models: In section 4, \Theta captures the set of parameters that completely define a given MDP model. We mainly introduced this parameterized representation of the model as we found it easier to describe the model reconciliation process in terms of these parameters than the elements of the standard model definition. Also, note that the function \Epsilon is defined as 2^{\Theta} --> \mathbb{M}, where \mathbb{M} is the space of possible models. We will update the section to further clarify the formulation.
>
> Need for ensuring optimality: This would clearly depend on the scenario at hand, for example in many mission-critical scenarios the user may actually care about the robot plans being optimal and hence would expect explanations that establish why the current actions were the optimal ones. This is a stance taken not only by works that follow explanation as model reconciliation but is also adopted by works like (Khan et al. 2009). Another advantage of ensuring optimality is that it could automatically account for all implicit foils, i.e. by providing enough information to ensure optimality, the explainer can ensure that there is enough information to refute any alternate plans the user may have had in their mind. With that said, while we use the concept of optimality in the theoretical sections, this doesn’t need to be reflected in the learned model. For example, if the user is happy with the plan being executable or being close to optimal, then they will label the transitions to be explicable when the explanations establish the minimal requirements.

---

> > ### Comment · AnonReviewer1 · 2019-05-15
> > **Why this plan?**
> >
> > Thanks for the response.
> >
> > While I agree that the general aim of XAIP is to answer 'why this plan?', I'm not convinced that model reconciliation answers 'Why this plan'? Pearl's work (see Book of Why or https://pdfs.semanticscholar.org/1756/88ed9003da8bc98c961d2e042dda27822252.pdf) identifies that answers to 'why?' questions refer to counterfactuals. Model reconciliations does not reason counterfactually. It answers questions on I think the first rung of the casual ladder, really just 'what is the model?'; or something similar. But it is more specific than that. Clearly articulating the question that model reconciliation answers would improve this paper. It gives some information for an observer to construct their own explanation for 'why this plan?', but it is not answering 'why this plan?' directly.

---

> > > ### Author Response · Authors · 2019-05-20
> > > **Re: why this plan?**
> > >
> > > Apologies for the late reply.
> > >
> > > We agree that “why” questions could involve performing some form of counterfactual reasoning, particularly contrasting the outcome of the current plan with possible alternatives (referred to as foils in contrastive explanation literature). Though we are not sure if the inferences specified in the context of causal inferences are the right point of comparison for the problems we are considering. Mainly because those recommendations are meant for cases where a user is querying for the answer when they agree with the system on the causal model. For example, if the user has a different causal model from the system, they may not agree with inferences made by the system. We would argue that in those cases, even when the user asks a why question, the system would need to first fix the user’s preconceived notions about the causal model before providing the results of inference. Model reconciliation is focused on providing the information corresponding to the first part. As mentioned before, we agree that for complex domains just providing model reconciliation information may not be enough, but may also need to provide inferential support (for eg: we can’t expect the user to calculate P(y_x |x ′ ,y ′ ) on their own).

---

> > > > ### Comment · AnonReviewer1 · 2019-05-20
> > > > **I agree!**
> > > >
> > > > Hi!
> > > >
> > > > I agree with what you say and note that counterfactual reasoning is not required to answer every question in explainable planning. What I mean is: your approach is answering a specific question; and that question is not a 'why' question. It is a 'what' question because it is giving factual information. 'What' questions can still be useful and are part of explanations, but clearly defining what the specific question is would help the reader of this paper.

---

### Official Review · AnonReviewer2 · 2019-05-13
**Model Reconciliation with learned model**

**Rating:** 4
**Confidence:** 3

**Review:**

This paper extends previous works from the same authors on model reconciliation.
The novelty here is that they drop the assumption about having the explicit user model. In this work, the model is instead learned.
This is clearly an interesting work, and it's worth presenting and discussing it at the workshop.

Some comments:
1) I noticed that the authors make a number of assumptions, but they are scattered throughout the paper. For example, some of things that the authors assume:
-the order in which the explanations are presented does not matter
-human and robot MDP models only differ on the transition probabilities
-the robot has access to a set of explanatory messages

and others. Perhaps it would be clearer to state from the beginning what the assumptions are, so that the reader do not discover them here and then?

2) I wonder how this approach can actually scale or be generalised to other (more complex) scenarios.
Again, I'm recommending acceptance of this paper, but I have to admit the use case considered is quite basic.
Can the authors comment on potential/limitation to scale to larger (and more complex/interesting) scenarios? Also, I think that the sentence "finding a complete policy explanation is relatively straightforward" should be put in context! As in many cases is not so straightforward!

3) What's the class of questions this approach aims to answer/address?
I didn't manage to understand which questions the authors are actually interested in.

4) User study
The user study looks nice (I did it myself through the website). But honestly I don't understand how you get from that an assessment of the effectiveness of the proposed explanations.

Minor/typos:
"another point point"
"standins"


In conclusion, I recommend accepting the paper. I have some major concerns, as I wrote, but I believe that XAIP being a new area we should be as inclusive as possible, and we should acknowledge that big contributions are made step by step.

---

### Decision · Program_Chairs · 2019-05-15

**Decision:**

Accept

**Comment:**

The reviewers agree to accept. Please address all review criticism as best possible for the final paper version and its presentation at the workshop. Looking forward to discuss your work at the workshop!